# The Response of Ecologically Functional Land to Changes in Urban Economic Growth and Transportation Construction in China

**DOI:** 10.3390/ijerph192114510

**Published:** 2022-11-04

**Authors:** Jingjing Liu, Jing Wang, Tianlin Zhai, Zehui Li

**Affiliations:** 1College of Water Sciences, Beijing Normal University, Beijing 100875, China; 2School of Resource and Environmental Sciences, Wuhan University, Wuhan 430079, China; 3College of Resources and Environmental Sciences, Henan Agricultural University, Zhengzhou 450002, China

**Keywords:** ecologically functional land, urban economic growth, transportation construction, spatiotemporal relationship, China

## Abstract

Understanding the impact of urban economic growth on ecologically functional land (EFL) change and the relevant mechanisms is necessary for adaptive ecological management and regional policy. The present study aims to explore the relationship between EFL change, urban economic growth and transportation construction based on reliable land survey data from 2000 and 2015, as well as natural and socio-economic data for over 2600 counties in China. We use the Two-Stage Least Squares (2SLS) technique to empirically analyze the temporal changes in their relationships and alleviate endogenous bias and use the Geographically Weighted Regression (GWR) model to explore the spatial heterogeneity across the country. The results indicate that the secondary and tertiary industries’ development had a significantly negative effect on EFL changes, and transportation construction is a major driver of urban economic growth in China, especially in the central region. From 2000 to 2015, the negative impact of urban economic growth on EFL changes decreased, and the contribution of transportation construction to urban economic growth increased. The regions (such as the central region) where transportation construction contributes more to the secondary and tertiary industries had a proportionally greater reduction in EFL. It appears that excessive dependence on transportation to drive the development of secondary and tertiary industries is the underlying reason for EFL reduction. The findings of this study can assist in formulating regional policies and advancing the coordination of urban economic development and ecosystem protection.

## 1. Introduction

As a basic natural resource, humans have utilized land in numerous ways for survival and development [1,2]. According to their major functions, land use types can be categorized into urban, agricultural, and ecological functional land (EFL) [3,4]. Along with an increasing population and rapid economic development, humans have continuously increased demands for natural resources, leading to land use changes [5,6]. Over the past few decades, China has experienced rapid industrialization and urbanization at the cost of encroachment on EFL and arable land [7]. Under the amplified effects of anthropogenic disturbance and environmental change, the destruction of EFL and degradation of ecosystems have increased over the past several decades [8]. The conflicts between ecosystem protection and economic development are getting worse [9,10,11]. Focusing on and quantifying the impacts of anthropogenic factors on environmental health is imperative to achieve sustainable regional and urban development.

Human well-being and sustainable development require ecological security, i.e., a stable and sustainable ecological environment [12]. To reverse the land degradation trend and promote ecological security, the Chinese government has implemented an array of ecosystem conservation and restoration projects [13]. In 2011, the Ecological Conservation Redline strategy was proposed to create an ecological protection pattern where the regions within the “red line” are strictly managed, and development is prohibited [14]. In the same year, the Chinese central government issued the National Main Functional Area Planning document, which designated key ecologically functional zones, major agricultural production zones, and optimized and prioritized development zones, according to their major function [15]. The optimized and prioritized development zones mainly referred to the urban agglomeration regions and metropolitan areas that support economic development. The major agricultural production zones primarily consist of agricultural land that has the primary function of food production. The key ecologically functional zones include various EFL types, such as forests, grasslands, wetlands, and water bodies, which have the main function of improving the environment and reducing ecosystem degradation [16]. The designation of these main functional areas is a significant step towards achieving sustainable development in China. In this context, understanding the response of EFL changes to the rapid development of the urban economy is important for coordinating the human–land relationship. 

Many studies have emphasized that distinct combinations of natural, political, and cultural drivers have a decisive influence on the landscape and land use changes [3,17,18]. Changes in EFL and ecosystem degradation in China have close relationships with socio-economic development [19,20,21]. The explanatory impact of socio-economic factors on EFL changes has gradually increased with economic development [22]. Some studies have explored the causes of changes in forest areas from the perspective of economic growth [23], management policies [24], environmental conditions, and armed conflicts [25]. Recent research has addressed the urbanization effects on ecosystem services [21,26]. These studies were mainly concerned with regional issues, and could have implications for cities, urban agglomerations, and parts of the country. Few studies have focused on EFL change and its response to economic growth on a national scale over time. This study aims to address this gap in the literature by exploring the relationship between EFL change and urban economic growth across the whole of China based on reliable and accurate land survey data from 2000 and 2015.

The problem of endogeneity is a widely discussed issue in the management literature, which may affect causal inferences and lead to spurious findings [27]. The ordinary least squares (OLS) estimation assumes that the independent variables are exogenous, i.e., not correlated with the error term [28]. Endogeneity bias occurs when an independent variable is correlated with the error term in the model [29]. The omission of variables, simultaneous causality, and errors in variables are the sources of endogeneity [30]. The instrumental variable (IV) method, especially the two-stage least squares (2SLS) technique, is a common and adaptable method to alleviate endogeneity problems [31]. One of the advantages of the 2SLS is that there is no restriction on variable distribution, but its disadvantage is that it requires a large sample size. The IV, a variable correlated with the endogenous independent variable but uncorrelated with the error term, affects the dependent variable indirectly through the endogenous independent variable [32]. The IV method decomposes the variations in the endogenous explanatory (independent) variable by using a valid IV to disregard the variations that bias the estimation, thus alleviating the endogeneity problem [33]. There is still a gap in knowledge regarding the consideration of the endogeneity problem in analyzing the relationship between EFL changes and urban economic growth.

China is experiencing rapid development of its economy and transportation facilities. Economic development is not only reflected in the optimization of the industrial structure, but also the improvement of public services and transportation systems [34]. In 2004, the state council launched the ‘mid-to-long-term railway development plan of China’, which was revised in 2008 and 2016, aiming to develop a high-speed rail network with a total length of 38,000 km by 2025 [35]. In 2022, the Ministry of Transport of the PRC announced the ‘outline of the mid-to-long-term development plan for scientific and technological innovation in the transportation sector (2021–2035)’ [36]. Transport infrastructure is considered a major driver of regional economic development, and any policy regarding transportation construction is an important regional and economic policy tool [37]. The transportation industry has an increasingly vital role in regional socio-economic development [38,39]. Land transportation, including highway, railway, and high-speed rail, has been the most significant sector in the transportation industry and has facilitated economic development [40]. From a small scale outlook (i.e., county and city), transportation construction may directly encroach on EFL and affect the ecological environment of the local area. From a long-term and macro perspective (i.e., national level), the coverture, operation and improvement of transport promote industrial and economic development by advancing interregional accessibility and reducing transaction time [41]. The development of transport, especially high-speed rail, tends to promote the increase of land use supply for logistics and warehouses and commercial and business use [42], thus affecting changes in non-urban land. Further economic growth has a feedback effect on transportation, increasing the demand for transportation and providing financial support to infrastructure construction [43]. With the continuous advancement of the transport policy, the driving effect of transport on the economy, especially the secondary and tertiary industries, is expected to increase. Therefore, when analyzing the relationship between EFL changes and urban economic growth, transportation construction satisfies the conditions for a suitable IV. The Geographically Weighted Regression (GWR) model is a spatial regression technique which can characterize spatial non-stationarity and obtain different regional regression coefficients by incorporating spatial dimensions into OLS regression [44,45]. We also used the GWR models to conduct the two stages of 2SLS regression to make the EFL change–urban economic growth relationship spatially explicit and the urban economic growth–transportation construction relationship clear.

An in-depth analysis of the response of EFL changes to urban economic growth and tackling the endogeneity problem empirically was needed to enrich the current literature and to provide proposals for improving the quality of the environment and life. Therefore, this study aimed to achieve three objectives: (1) to quantify the impact of urban economic growth on EFL changes over time, using correlation analysis and the OLS method; (2) to explore the influencing mechanism of urban economic growth on EFL changes using 2SLS, choosing transportation construction as the IV for urban economic growth (3) to identify and compare the spatial patterns of the EFL change–urban economic growth relationship and the urban economic growth–transportation construction relationship using the GWR method. Robust findings from this study can provide scientific evidence for formulating regional policies and promoting ecological security in China and fills a gap in the literature on interactions between environmental systems and human activities, specifically ecologically functional land and urban economic growth relationship research. 

This paper proposed the following hypotheses to advance our investigation:

**Hypothesis** **1.**Urban economic growth will reduce ecologically functional land, and the effects will have spatial heterogeneity across different regions of China.

**Hypothesis** **2.**Transport’s contribution to urban economic growth will significantly increase from 2000 to 2015 and will show regional heterogeneity.

**Hypothesis** **3.**Reliance on transport to drive urban economic growth may harm ecologically functional land.

## 2. Materials and Methods

### 2.1. Data and Variables

#### 2.1.1. Explained Variables

The land-use data was sourced from the series of national land surveys and annual land-use change surveys, which are the most legally effective and comprehensive surveys in China. Based on digital ortho-photo maps, the land and land use change survey database was aided by available maps and field surveys covering nearly 3000 counties [3,46]. The third national land survey was concluded in 2021, and the national dataset has not yet been made public. The latest available data is from 2015, thus, the national land-use data at the county level in 2000 and 2015 was chosen and processed to carry out our study. These two years of land use classification were unified into 12 first class and 56 s class (the land classification of the second national land survey GBT21010-2007). In the National Main Functional Area Planning document in 2011, the territory of China was designated into different functional areas according to their dominant functions, such as ecological and food production functions [15]. Ecologically functional land (EFL) refers to the spatial land units with the dominant function of providing ecological services and maintaining key ecological processes. Thus, the EFL includes water bodies, wetlands, glaciers and snow, saline land, forest land, shrubland, natural grassland and other grasslands. [3,47]. The proportion of EFL to total territorial space was calculated as the dependent variable of this study. Figure 1 shows the spatial distribution of the EFL proportion in 2000 and 2015. The overall distribution of EFL was high in the west and low in the east and showed no significant change from 2000 to 2015. The central and central-east regions, such as the provinces of Henan, Shandong, Shaanxi, Shanxi, Sichuan, Jiangsu, and Hebei, are typical low-value regions. The proportion of EFL in the Shanxi and Shaanxi Provinces increased from 2000 to 2015. 

#### 2.1.2. Core Explanatory Variables

Urban economic growth is the core explanatory variable. According to most of the current studies, economic density, i.e., regional Gross Domestic Product (GDP) per square kilometer, was chosen as the proxy variable of economic growth. The secondary and tertiary industries are the main sectors of the urban economy [35]. Data on the secondary and tertiary industries’ GDP in 2000 and 2015 was used to calculate the urban economic density (ED), collected from the “China Statistics Yearbook”. 

In many countries, economic growth is anchored to transportation. In the short term, transport infrastructure increases construction enterprise numbers and job opportunities in construction [48]. In the long term, investment in transportation can increase the size of the agglomeration and the magnitude of external economies [49] and positively affects economic growth and the corresponding regions’ development [50]. Transport improvements could serve as a better household amenity and lure in migrants [51], as well as foster the movement of goods and services considerably [42]. In China, transport infrastructure has an important role in regional economic growth and has clear spatial spillovers [52]. Land transport is the dominant mode in China and has a greater impact on regional economic growth than other transport modes [53]. Driving economic growth through transportation is one of China’s important strategies since transportation is a pivotal element in the selection of industrial locations [44]. As the Chinese government vigorously promotes transportation construction, its role in promoting economic development and economic coordination in the eastern, central and western regions is expected to become more prominent. Therefore, we considered land transport as an IV for urban economic growth to control endogeneity. Road density (RD), which generally reflects the level of traffic line development [54], was chosen as a proxy variable of transportation construction. The data on roads, collected from the annual Chinese national survey on land-use change, include railways and highways data, for calculating the road density.

#### 2.1.3. Control Variables

To control for the omitted variable bias, we also used several control variables. Elevation and slope are the general natural factors that limit the distribution of EFL [55]. Areas with flatter and lower elevations are more susceptible to development into cultivated and built-up land. Climate factors, such as precipitation and temperature, also strongly correlate with vegetation coverage, land ecosystems, and wetland [56,57,58]. Therefore, DEM (Digital Elevation Model) data, average annual precipitation, and annual temperature data in 2000 and 2015 were chosen as geographical control variables, which were collected from the Resources and environment data cloud platform (http://www.gscloud.cn (accessed on 23 May 2022)). Under the joint effects of geographical characteristics and human activities, there is no doubt that industrial use and agricultural expansion have greatly impacted environmental conservation and ecological land space [55,59], and population density has been proven to be the main driver of EFL evolution [60]. Regions with dense populations are likely to meet the demands for agricultural production and economic benefits by occupying a larger amount of EFL [19]. Considering the difficulty of obtaining data on industrial use and agricultural expansion for more than 2600 counties, we chose population density as a proxy variable for human activities. Data on the urban population was used to calculate the urban population density in 2000 and 2015 and was cited from the “China Statistics Yearbook”. 

Table 1 shows the descriptive statistics of all variables. The values of the standard deviation showed that precipitation, temperature, and elevation registered significant volatility, indicating the huge differences in geographic characteristics across China. The Jarque–Bera test indicated that all the chosen variables were non-normally distributed.

### 2.2. Model Settings

We transformed all the variables in our models with natural logarithms to get a more stable data series and eliminate the heteroscedasticity and multicollinearity of the models. As a reference for the 2SLS model and GWR model, the following benchmark regression was constructed by the OLS model first:(1)$${\ln\mathrm{EL}}_{i}{=\alpha }_{0}{+\alpha }_{1}{\ln\mathrm{ED}}_{i}{+a}_{2}{\ln X}_{i}{+u}_{i}{+\varepsilon }_{i}$$
where *i* indicates the county; *EL* is the proportion of EFL, *ED* denotes urban economic density; *X* represents other independent variables, including urban population density, elevation, and control variables, including temperature and precipitation; *α*_0_ is a constant term; *α*_1_ denotes the impact coefficient of urban economic development on the proportion of EFL; *u_i_* is the fixed effects of the county, and *ε_i_* is the error term.

To investigate the relationship between EFL, urban economic growth, and transportation construction, the 2SLS model was set as Equations (2) and (3). This can confirm the robustness of the empirical results better. The variables used in the 2SLS model were the same as in the OLS model.
(2)$${\ln\mathrm{ED}}_{i}{=\beta }_{0}{+\beta }_{1}{\ln\mathrm{RD}}_{i}{+\beta }_{2}{\ln X}_{i}{+\delta }_{i}{+\tau }_{i}$$
(3)$${\ln\mathrm{EL}}_{i\;}{=\gamma }_{0}{+\gamma }_{1}{\ln\mathrm{ED}}_{i}{+\gamma }_{2}{\ln X}_{i}{+u}_{i}{+\varepsilon }_{i}$$
where road density is defined as *RD*, which is the instrumental variable of urban economic density; *ED* denotes urban economic density; *EL* represents the proportion of EFL; *X* represents other independent variables and control variables; *β*_0_ and *γ*_0_ are constants; and *β*_1_, *β*_2_, *γ*_1_ and *γ*_2_ are estimated coefficients. *δ_i_* and *u_i_* represent the fixed effects of the county. *τ_i_* and *ε_i_* are the disturbance terms. The evaluations of the OLS and 2SLS models were completed in STATA 17.

The unbalanced distribution of natural and socio-economic factors in different regions creates interregional spatial heterogeneity. Therefore, global parameters cannot explain spatial heterogeneity. The GWR model, an improved traditional linear regression model, is an effective method for exploring spatial heterogeneity. It considers spatial heterogeneity and utilizes geographic coordinates and core functions to perform local regression estimations on adjacent individuals in each group [61]. The GWR model is expressed as follows:(4)$$Y_{i}{=\beta }_{0}\left(u_{i},v_{i}\right)+\overset{p}{\underset{k=1}{\sum }}\beta_{k}\left(u_{i},v_{i}\right)X_{\mathrm{ik}}{+\varepsilon }_{i}$$
where *i* represents the individual sample, *Y* denotes the dependent variables, *X* is the normalized affect factors, *k* is the total number of grid cells, and *ε_i_* is the random error. (*u_i_*, *v_i_*) denotes the spatial location of the sample *i*, *β*_0_(*u_i_*, *v_i_*) denotes the intercept constant of sample *i*; and *β_k_*(*u_i_*, *v_i_*) denotes the coefficient of the *k*th spatial variable of sample *i*. They were estimated by the local weighted least squares method. The estimations of the GWR models were completed in ArcGIS 10.2. The optimal bandwidth was set by the minimum Akaike Information Criterion (AIC).

## 3. Results

### 3.1. Correlation Analysis

Correlation analysis is fundamental before running an econometric estimation [62,63]. We conducted correlation analysis and computed Pearson coefficients to check for potential multicollinearity issues regarding the independent variables in 2000 and 2015. Figure 2 shows the *r* values among different variables. Urban economic growth had a negative correlation with ecologically functional land in 2000 and 2015, with coefficients of −0.54 (*p* < 0.01) and −0.59 (*p* < 0.01), respectively. Figure 3 gives a spatial correlation between EFL proportion and economic density using the bivariate spatial autocorrelation tool in GeoDa. Their spatial correlation showed High(EFL)–Low(ED) in the west part and Low(EFL)–High(ED) in the east part, indicating a significant trade-off relationship. Only very few counties in the east and south regions had a High(EFL)–Low(ED) relationship. The variance inflation factor (VIF) tests were performed to verify multicollinearity. The results (Table 2) showed the VIF values of all variables in 2000 and 2015 were below 5, indicating a low risk of multicollinearity. 

### 3.2. Impacts of Urban Economic Growth on Changes in EFL in China

For reference, the OLS regression results from 2000 and 2015 are listed in Table 3. From model (1) to model (3), the control variables were included one by one. The estimated coefficients of urban economic density (ln*ED*) showed no significant change, indicating that urban economic density had a negative impact on the proportion of EFL (ln*EL*), and the results were robust. Specifically, according to the coefficients of model (3), raising urban economic density by 1% would lead to a reduced EFL proportion of 0.239% in 2000 and 0.16% in 2015. These results supported hypothesis 1. Compared to the coefficients of other variables, urban economic density had the largest impact on the EFL in 2000. By 2015, the impact of urban economic density on the EFL was slightly lower than that of elevation. Additionally, the estimated coefficients of the control variables also had practical significance. Urban population density (ln*PD*) had a positive impact on EFL in 2000, but this turned into a negative effect in 2015. The increasing population density led to a large amount of ecologically functional land being occupied to satisfy demands for living and production space. For natural factors, there was a clear positive correlation between elevation (ln*DEM*) and the proportion of EFL. The results were consistent with the finding that topography plays a significant role in ecological land [64]. The impact of temperature and precipitation on the proportion of EFL was relatively small.

### 3.3. Relationship among Transportation Construction, Urban Economic Growth and Changes in EFL

The 2SLS regression was used to alleviate endogeneity problems in our models, as well as to analyze the relationship between transportation construction, urban economic growth, and the proportion of EFL. Table 4 reports the regression results of the 2SLS. Column (1) and column (2) are the second-stage regression results and the first-stage regression results, respectively. The results of the Durbin–Wu–Hausman test supported our models’ decision to treat variables as endogenous. The first-stage regression results demonstrated that the coefficients of road density (ln*RD*) were positive at the 1% significance level, combined with the F value, which revealed that the chosen instrumental variable had a strong explanatory power for urban economic density. The road density increased by 1%, leading to a 0.694% increase in urban economic density in 2000 and a 1.4431% increase by 2015.

For the second-stage regression results, the estimated coefficients of urban economic density (ln*ED*) were negative at the 1% significance level. The negative impact of urban economic growth on the proportion of EFL in 2015 decreased compared to 2000. The urban economic density increased every 1%, leading to a 0.315% reduction in EFL proportion in 2000, and a 0.193% reduction by 2015. The absolute value of this was greater than the OLS regression results, indicating that endogenous problems may lead to underestimating the impact of urban economic growth. The coefficients of other control variables were consistent with the OLS regression results, which verified that the instrumental variable regression was robust. The first stage and second-stage regression results of the 2SLS implied that transportation construction has played an increasingly important role in urban economic growth. The results supported hypothesis 2. In the following section, we will use the GWR models to analyze their spatial distribution patterns and further explore the spatiotemporal relationship among those three variables.

### 3.4. Spatial and Temporal Variation Analysis of the Response of Changes in EFL to Urban Economic Growth and Transportation Construction

The diagnosability of the OLS and 2SLS models indicated that the urban economic density, population density, and elevation were the main factors influencing the proportion of EFL. The GWR model was used to analyze the spatial variation of EFL, and urban economic density based on the consideration of those key influencing factors. Another GWR model based on Equation (2) was also applied to analyze the spatial variation of the impact of transportation construction on urban economic growth. By comparing the EFL–ED relationship and ED–RD relationship, we tried to analyze their spatial consistency and differences. Considering that the explanatory variables followed an approximately normal distribution in the GWR, normality tests were performed before building the GWR models. The final results denoted that the variables, including the proportion of EFL, elevation, urban economic density, population density and road density, followed an approximately normal distribution using the histogram with a normal fit line in STATA 17. 

Table 5 lists the parameters in the two GWR models, showing that the goodness-of-fit values of the models were all above 0.5, and higher than that in the OLS and 2SLS models. Figure 4 and Figure 5 show the spatial heterogeneity of the two GWR models in terms of county-level fitting degree, which was reflected in the spatial variation of local R^2^ in 2000 and 2015. Both the local R^2^ in the model with the proportion of EFL (ln*EL*) as the dependent variable and the model with urban economic density (ln*ED*) as the dependent variable were greater than the corresponding global R^2^. This suggested that the GWR models were superior to global regression models in terms of interpreting changes in EFL and urban economic growth in China at both the global and local levels.

As shown in Figure 6, the local coefficients of ln*ED* were negative in 2000 and 2015 and demonstrated significant regional heterogeneity. In 2000, a third of counties had coefficients between −0.32 and −0.13. By 2015, more than 65% of the counties had coefficients between −0.32 and −0.13. Counties with a coefficient lower than −0.32 (absolute value higher than 0.32) decreased significantly from 2000 to 2015. Looking into the different regions, the negative impact of urban economic growth on EFL proportion in the undeveloped western region was smaller than that in the central and eastern regions. The central provinces, dominated by secondary industries, including the provinces of Shanxi, Henan, and Hebei, showed the largest negative effect in 2000. Along with the implementation of a series of ecosystem protection programs, especially the Grain for Green program aimed at transforming croplands with steep slopes into forests and grasslands from 1999, the counties with coefficients lower than −0.89 were reduced by 2015; they were only distributed in a small part of Henan and Hebei provinces. The response degrees of the proportion of EFL to economic growth in Xinjiang and Inner Mongolia Autonomous Region also clearly declined from 2000 to 2015. The response degrees increased in a small part of some provinces, such as the Tibetan Autonomous Region and Guangdong province, from 2000 to 2015, which was directly related to the anabatic ecosystem degradation in these areas [4]. Although the negative effect decreased from 2000 to 2015, urban economic growth still had a more significant negative impact on the proportion of EFL in the central region. The results further confirmed our hypothesis 1.

As displayed in Figure 7, the spatial distribution of local coefficients for road density (ln*RD*) showed significant spatial variation in 2000 and 2015. The contribution of transportation construction to regional economic growth varied across regions and depended on the economic development level. Over 45% of the counties had coefficients above 0.45 in 2000, which reached over 75% by 2015, indicating the increasing impact of transportation construction on urban economic growth in China. The central and southwest regions exhibited significant spatial aggregation in 2000; for every unit (1%) of the increase in road density, the average economic density increased by over 0.75%. By 2015, the counties with coefficients above 0.75 had expanded significantly around the central southwest regions. The study of Chen [65] indicated that investment in transport in the southwest region had a more significant effect on economic growth than that in developed eastern regions. By comparing Figure 3 and Figure 4, the higher coefficients for ln*RD* partially overlapped with the distribution of the areas where EFL was greatly affected by urban economic growth, especially in Inner Mongolia, Heilongjiang, Henan, and Shaanxi provinces. In addition, in provinces such as Jiangsu, Guangdong, and Guangxi, the contribution of transportation construction to economic growth increased, and the negative effect of economic growth on EFL changes also increased. The areas where transportation contributed more to economic growth had a more significant negative impact on EFL changes. The results supported hypothesis 3.

## 4. Discussion

Since the economic reforms in 1978, China has undergone a tremendous change inland-use and economic development. The secondary and tertiary industries were the main sectors driving urban economic development, which brought about GDP growth, but reduced the proportion of EFL. According to the Kuznets curve [66], when the socio-economic level is low, slow economic growth will not have a significant effect on ecosystems; when economic development reaches a high level, the negative effect will weaken. Only a medium level of economic development dramatically affects ecosystems [67]. Correspondingly, we observed the largest negative impact of urban economic growth on EFL proportion in the central region, the development level of which is between that of the west and the east, and the lowest negative impact was observed in the undeveloped western region. The effect of economic growth on EFL and ecosystem services is dynamic, depending on changing socio-economic levels and development phases [68]. Consistent with the gradient transfer theory [69], in the period studied, central China was in a period of accelerating industrialization, had high proportions of the primary and secondary sectors, and had low economic resilience [70]. A development mode that emphasizes industrial structure optimization and development quality is needed for ecosystem protection and sustainable development.

Driving economic growth through transportation is one of China’s most important strategies. As the Chinese government has launched a series of transport planning and policies, the transport network has been rapidly extended [71]. Provinces such as Shaanxi and Shanxi, located in the central Longhai–Lanxin Economic Belt, were key areas of the “Western Development” strategy. The transportation infrastructure of those regions was conducive to the development of local secondary and tertiary industries and the economies in the western and central regions [39]. However, relying on transport to drive economic growth and narrow the gap between the east and the west might harm ecological land space and ecosystems. In addition, the goal of establishing the transportation development strategy was mainly to promote social and economic development and rarely involved protecting environmental benefits [72]. The suitability and effectiveness of transport should be highlighted in planning and practice.

In this study, we used the 2SLS and GWR models to assess the impact of urban economic growth on the proportion of ecologically functional land (EFL) in 2000 and 2015. The results of different regression models identified the temporal and spatial differences in the effects and the robustness of our results. Although this study provides a perspective by which to analyze the response of changes in EFL to economic growth and transportation construction, the interactions among them may be more complicated than our results indicate. This study analyzed the effect of road density, including highways and railways, on economic growth, but the impacts of transportation on industrial development and economic growth vary with different types of industries [73]. The impact of the specific transportation mode, including high-speed rail, should be considered in future studies. Given that the transportation construction of China is still in the process of rapid development, it would be more valuable to explore this using multiple years of data in the regressions to provide a more comprehensive analysis over a longer time horizon. 

## 5. Conclusions

This study explored the response of EFL changes to urban economic growth in over 2600 counties in China. The spatiotemporal relationship among EFL changes, urban economic growth, and transportation construction was explored using the 2SLS and GWR models. The overall negative impact of secondary and tertiary industry development on EFL declined from 2000 to 2015. Urban economic growth in the central region, which has a development level between the undeveloped western region and the developed eastern region, had the largest impact on the reduction in EFL. Transportation construction was very useful in promoting industrial and urban economic development, especially in the central region, including the provinces of Shaanxi, Shanxi and Hebei. The impact of urban economic growth may change as economic development reaches different stages and through industrial structure optimization. The development mode relies on transport, and regional policies emphasize the priority of transportation construction; however, this should be improved. This study advances our understanding of the mechanism of urban economic growth affecting EFL changes and provides evidence for policymaking and science-based transportation development planning.

## Figures and Tables

**Figure 1 ijerph-19-14510-f001:**
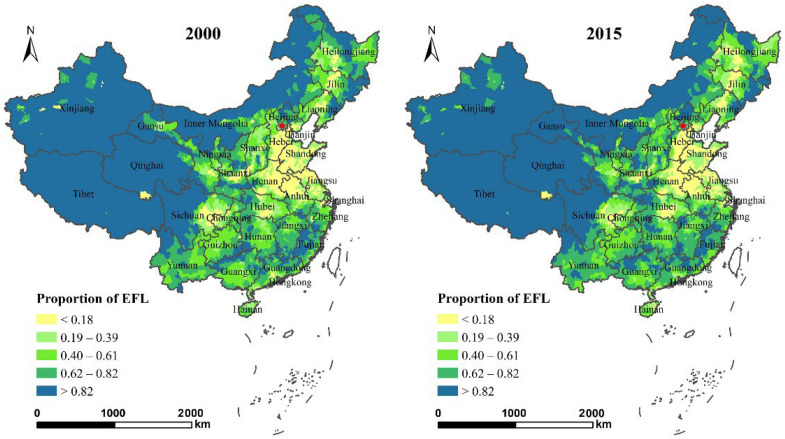
Proportion of ecologically functional land (EFL) to total territorial space in 2000 and 2015.

**Figure 2 ijerph-19-14510-f002:**
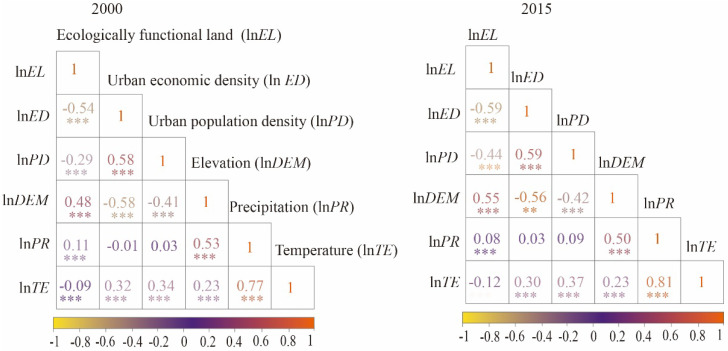
Correlations among ecologically functional land (ln*EL*), urban economic density (ln*ED*), and control variables. ** and *** denote significance at the 5% and 1% levels, respectively.

**Figure 3 ijerph-19-14510-f003:**
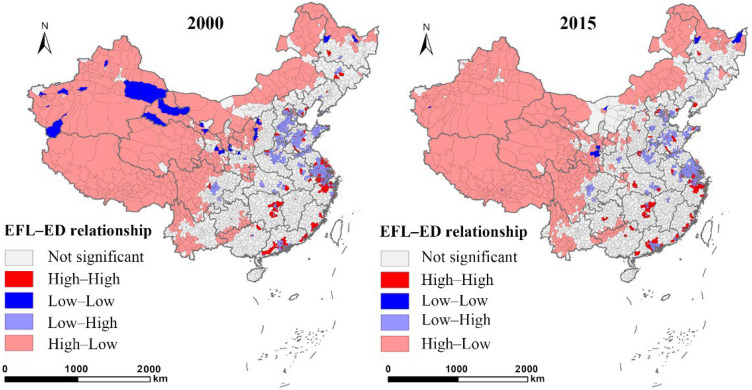
Spatial correlation between EFL proportion and urban economic density (ED) in 2000 and 2015.

**Figure 4 ijerph-19-14510-f004:**
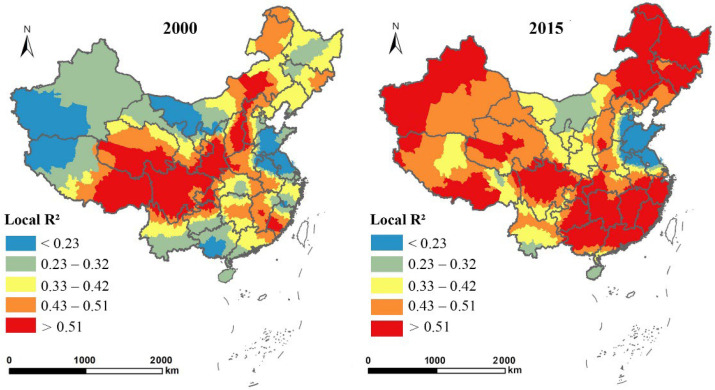
Spatial distribution of local R^2^ in the GWR model with the proportion of EFL (ln*EL*) as the dependent variable in 2000 and 2015.

**Figure 5 ijerph-19-14510-f005:**
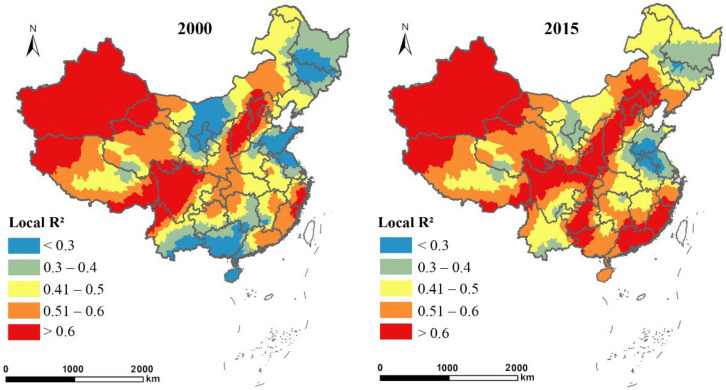
Spatial distribution of local R^2^ in the GWR model with urban economic density (ln*ED*) as the dependent variable in 2000 and 2015.

**Figure 6 ijerph-19-14510-f006:**
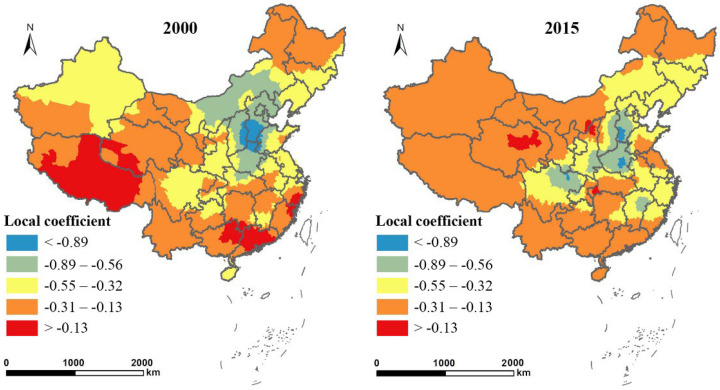
Spatial distribution of the local coefficients of urban economic density (ln*ED*) in 2000 and 2015.

**Figure 7 ijerph-19-14510-f007:**
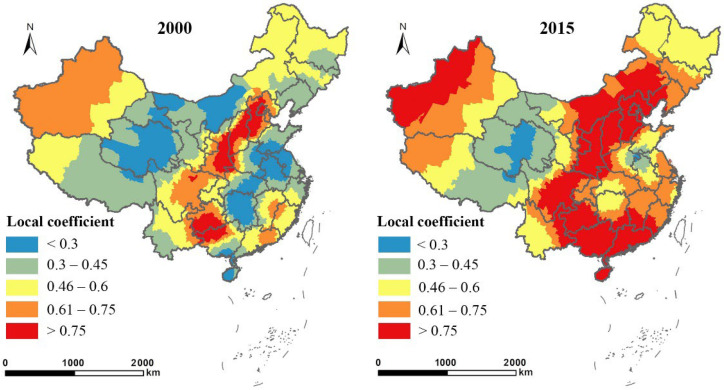
Spatial distribution of the local coefficients of road density (ln*RD*) in 2000 and 2015.

**Table 1 ijerph-19-14510-t001:** Descriptive statistics of all the variables.

Variable	Sources or Methods	Abbr.	Year	Mean	Max	Min	Std. Dev.	Skewness	Kurtosis	Jarque–Bera Test	Probability
The proportion of ecologically functional land (EFL) (*n* = 2607)	Percentage of EFL in all land use	EL	2000	0.418	0.999	0	0.29	0.178	1.735	187.6	0.001
			2015	0.506	0.998	0	0.29	−0.093	1.762	170.3	0.001
Urban economic density (*n* = 2607)	Regional GDP of the secondary and tertiary industries per km^2^ (10 × 10^7^ yuan /km^2^)	ED	2000	0.045	2.15	0	0.12	7.742	88.06	810,165	0.00
			2015	0.449	25.18	0	1.47	9.202	118.80	1,500,046	0.00
Road density (*n* = 2607)	Length of the traffic line per square kilometer (km/km^2^)	RD	2000	0.013	0.202	0	0.013	2.775	25.531	58,004	0.00
			2015	0.022	0.259	0	0.018	2.375	17.240	24,004	0.00
Urban population density (*n* = 2607)	Regional urban population density (100 persons/ km^2^)	PD	2000	44.08	461	0	34.73	2.357	19.104	31,434	0.00
			2015	50.49	547.49	0	40.58	2.897	24.098	52,064	0.00
Elevation (*n* = 2607)	Digital Elevation Model data (m)	DEM	2000	761.91	5146.59	0	1039.29	2.241	8.235	5159	0.00
			2015	761.91	5146.59	0	1039.29	2.241	8.234	5159	0.00
Precipitation (*n* = 2607)	Average annual precipitation (0.1 mm)	PR	2000	9191.65	30,027.14	0	5576.48	0.369	2.471	89.55	0.001
			2015	9237.62	25789.8	0	6324.17	0.586	2.356	194.5	0.004
Temperature (*n* = 2607)	Average annual temperature (0.1 °C)	TE	2000	121.49	253.66	−64.21	65.14	−0.534	2.393	163.8	0.002
			2015	123.57	264.28	−43.20	64.55	−0.567	2.351	185.4	0.00

**Table 2 ijerph-19-14510-t002:** Results of the variance inflation factor (VIF).

Variable	ln*ED*	ln*PD*	ln*DEM*	ln*TE*	ln*PR*
VIF (2000)	2.43	2.41	2.16	4.14	4.32
VIF (2015)	2.98	2.59	2.37	3.47	3.93

Note: ED = Economic density; PD = Population Density; DEM = Elevation; TE = Temperature; PR = Precipitation.

**Table 3 ijerph-19-14510-t003:** Parameter estimation of OLS regression in 2000 and 2015.

Variable	(1)		(2)		(3)	
	2000	2015	2000	2015	2000	2015
ln*ED*	−0.255 *** (0.013)	−0.169 *** (0.01)	−0.239 *** (0.015)	−0.157 *** (0.011)	−0.239 *** (0.015)	−0.160 *** (0.011)
ln*PD*	0.087 *** (0.026)	−0.106 *** (0.02)	0.107 *** (0.027)	−0.086 *** (0.0212)	0.105 *** (0.027)	−0.096 *** (0.021)
ln*DEM*	0.139 *** (0.011)	0.141 *** (0.008)	0.159 *** (0.014)	0.156 *** (0.01)	0.164 *** (0.015)	0.172 *** (0.012)
ln*TE*			−0.043 ** (0.017)	−0.035 ** (0.0129)	−0.028 (0.025)	0.034 * (0.021)
ln*PR*					−0.014 (0.017)	−0.058 *** (0.014)
Observations	2582	2602	2582	2602	2582	2602
R^2^	0.333	0.427	0.335	0.429	0.335	0.433

Note: The standard errors are in parentheses; *, **, and *** denote significance at 10%, 5% and 1%, respectively. ED = Economic density; PD = Population Density; DEM = Elevation; TE = Temperature; PR = Precipitation.

**Table 4 ijerph-19-14510-t004:** Parameter estimation of the 2SLS regression in 2000 and 2015.

Variable	(1) ln*EL*		(2) ln*ED*	
	2000	2015	2000	2015
ln*ED*	−0.315 *** (0.021)	−0.193 *** (0.024)		
ln*RD*			0.694 *** (0.026)	1.443 *** (0.024)
ln*PD*	0.507 ** (0.160)	0.494 (0.061)	0.165 *** (0.033)	0.063 *** (0.03)
ln*DEM*	0.14 *** (0.049)	0.107 *** (0.02)	−0.409 *** (0.016)	−0.141 *** (0.01)
ln*TE*	−0.117 (0.028)	0.033 *** (0.029)	0.033 ** (0.018)	0.081 ** (0.02)
ln*PR*	0.03 (0.023)	−0.035 ** (0.019)	0.043 *** (0.013)	0.038 *** (0.014)
R^2^	0.372	0.3818	0.674	0.763
Observations	2582	2582	2602	2602
Wu-Hausman F			19.2306	14.5684
*p*-value			0.012	0.007
F value			1069.81	1677.36

Note: The standard errors are in parentheses; ** and *** denote significance at 5% and 1%, respectively. ED = Economic density; PD = Population Density; DEM = Elevation; TE = Temperature; PR = Precipitation.

**Table 5 ijerph-19-14510-t005:** Estimation parameters of GWR models.

Variables	Year	Parameters				
		Adjusted R^2^	Residual Squares	Effective Number	Sigma	AICc
The proportion of ecologically functional land (ln*EL*)	2000	0.565	1049.606	173.428	0.623	5540.953
	2015	0.695	770.208	177.856	0.534	4657.103
Urban economic density (ln*ED*)	2000	0.751	3100.486	154.284	1.066	8631.551
	2015	0.845	3333.34	149.34	1.105	8832.623

## Data Availability

The datasets analyzed during the current study are available from the corresponding author on reasonable request.

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
