# Peer review of "The Response of Ecologically Functional Land to Changes in Urban Economic Growth and Transportation Construction in China"

_ijerph, 2022, doi:10.3390/ijerph192114510_

Round 1

Reviewer 1 Report

Review of the Manuscript Manuscript ijerph 1918599  Exploring the Response of Changes in Ecologically Functional Land to Urban Economic Growth and Transportation Construction in China” for the International Journal of Environmental Research and Public Health.

General Comments

From my point of view, it is a very interesting topic and simultaneously it seems that to the best of my knowledge is the first empirical research which study the role of the Exploring the Response of Changes in Ecologically Functional Land to Urban Economic Growth and Transportation Construction in China, by allowing economic systems to continue operating partially.

The paper consists of the following sections: Introduction, Theories of development zones, Study area, Materials and Methods, Results, Discussion and Conclusions.

However, I find some recommendations:

1.       The abstract must contain the main purpose of the paper, the research method used in the research and the main contributions.

2.       It would be very useful to add in the "Introduction" section the purpose, objectives and hypothesis of the research.

3.       I recommend the authors to make a complete descriptive analysis and to include a series of indicators and tests such as standard deviation, Jarque-Berra, Kurtosis, probabilities, etc., and the number of observations taken in the sample.

4.       It is very important that the authors present the correlation matrix and the covariance matrix and explain the results obtained.

5. In the econometric section the authors have to apply the fixed effect estimation or the random effect estimation (see for instance, Baltagi (2008), Hsiao (2014) and AndreB et al. (2015)). Besides, the corresponding tests to determine which is the best method of estimation is needed (see the Hausman test, the Breusch and Pagan (1980)´s Lagrange multiplier, the F test for fixed effects to test whether all unobservable individual effects are zero).

6. Additionally, they should test whether there is endogeneity because this problem must be taking into account in the estimation methodology.

7. The authors talk about the long-run relationship between these variables, however they do not support the empirical evidence providing panel cointegration tests that are crucial (see for instance Kao (1999) panel data cointegration test, the Pedroni (1999, 2004) panel data cointegration test or the Westerlund (2005) panel data cointegration test, among others).

8. It should be interesting that the authors estimate the marginal effects before and after the financial crises to compare before and after this particular structural break.

9. I think that descriptive perspective is crucial to understand the context of the problem to be analyzed.

10. We consider that VIF test is very important for eliminating multicollinearity problems.

11.    Also,  we consider the literature is not enough and that is why, we recommend the authors to refer to other recent works indexed in Web of Science, Scopus, Emerald, Cambrige, and of course MDPI Journals. We suggest that the authors cite papers published in MDPI journals and Web of Science Journals, such as:

  1. Batrancea, L. (2021) The Influence of Liquidity and Solvency on Performance within the Healthcare Industry: Evidence from Publicly Listed Companies,  Mathematics 9, no. 18: 2231. https://doi.org/10.3390/math9182231, eISSN:2227-7390/sept.2021
  2. Batrancea, L.; Rus, M.I.; Masca, E.S.; Morar, I.D. Fiscal Pressure as a Trigger of Financial Performance for the Energy Industry: An Empirical Investigation across a 16-Year Period. Energies 202114, 3769. https://doi.org/10.3390/en14133769

All in all, I consider that the paper must be improved. As a result, the article can be published in the prestigious International Journal of Environmental Research and Public Health after minor revisions.

Author Response

Thank you very much for taking the time to read this manuscript and make valuable comments and suggestions.  Please see the attachment for our response.

We appreciate this opportunity for revising and improving our manuscript and made a careful and thorough revision according to your valuable and reasonable comments. 

Author Response

(The authors gave the same response as above.)

Reviewer 3 Report

comments on this mansucript were included in the revision file

Author Response

Thank you very much for taking the time to read this manuscript and make valuable comments.  Please see the attachment for our response.

We appreciate this opportunity for revising and improving our manuscript and made a careful and thorough revision according to your valuable and reasonable comments. 

Round 2

Reviewer 2 Report

Please see document attached.

Author Response

Dear reviewer,

We have revised our manuscript carefully according to your comments, please see the attachment

Reviewer 3 Report

This manuscript has been greatly improved. The authors have correctly interpreted the reviewers' comments. However, I consider that it does not yet have sufficient technical merit to be published in the journal. 

I believe that the time series database should be expanded, or at least updated, and less traditional spatial analysis techniques should be applied

Author Response

Dear reviewer,

Many thanks for your valuable comments on the manuscript. I believe that expanding the time series database and using more advanced methods could enrich the manuscript and make it more popular. However, in a country as large as China (nearly 3000 counties), for now, it is very hard to obtain updated land survey data because it often takes several years for national data to be processed and made public. Acquiring other socioeconomic data at the county level is also incredibly laborious. 
We really want to further enrich our manuscript. We have tried to use remote sensing data of land use (including 2020) to expand and update our database, and we also would like to spend more time supplementing socioeconomic data. We compared remote sensing data with land survey data, they are quite inconsistent. Remote sensing data can indeed reflect the land use situation, but the series of national land surveys dataset is the most reliable land use data in China. The error between these two different sources of data will lead to inaccurate results. Moreover, although we can obtain the above data, road density data is still not available because it cannot be derived from remote sensing data.
It is true that using less traditional spatial analysis techniques may make the article more popular and meaty. But, we have tried different methods and settled on the one that was more suitable and had robust results, and the robustness of the results in the article has been demonstrated in several ways.  We appreciate your valuable suggestions, and we are willing to try and improve in our future research.